# Comparison of the Properties of a Random Copolymer and a Molten Blend PA6/PA6.9

**DOI:** 10.3390/polym14194115

**Published:** 2022-10-01

**Authors:** Maddalena Bertolla, Michele Cecchetto, Mattia Comotto, Anacleto Dal Moro

**Affiliations:** Aquafil S.p.A., Via Linfano 9, 38062 Arco, Italy

**Keywords:** PA6.9-ran-PA6 (PA6.9-6), copolymerization, randomness

## Abstract

This study compares the thermal and mechanical properties of two different materials, obtained via two diverse synthetic pathways. The first one is a mixed blend of PA6/PA6.9, while the second is a random copolymer (PA6.9-ran–PA6, obtained via copolymerization of its monomers, i.e., caprolactam, hexamethylenediamine and azelaic acid). Several tests are carried out according to the aforementioned pathways, varying the relative ratio between the two polymeric building blocks. The role of the synthetized plastic is to be coupled to polyamide material, such as PA6, to confer its better properties. The synthetized random copolymer, besides displaying ease of processability with respect to conventional methods, exhibits interesting features. It has a low melting point (135 °C, PA6.9-ran-PA6 50:50) and therefore it might be used as a hot-melt adhesive in composite material. Owing to its low crystallinity content, the material displays a rubber-like behavior and may be employed to confer elastomeric properties to PA6 matrix, in place of non-amidic material (for example elastomeric polyurethanes). This leads to a further advantage in terms of chemical recyclability of the end-of-life material, since the additive increases the percentage of PA6 in waste material and, consequently, the yield of caprolactam recovery.

## 1. Introduction

Plastics are widely used in daily lives as packaging, clothing and sports equipment, flooring sector, biomedical devices, electronic components and many other applications. The demand for plastics has increased over the years, while most of them derive from nonrenewable feedstocks. In addition, the presence of post-use plastics in landfills, rivers and oceans triggers a big environmental concern and results in economical loss. Given these reasons, the main efforts in the plastics field are related to the improvement and study of their recovery, separation and recycling [1,2].

Not all plastics can be easily recycled, and according to the polymer type, different recovery processes can be evaluated. The polymer considered of interest in this study, PA6, has the advantage of being chemically recyclable. In other words, via hydrolytic depolymerization, the polymeric chains of PA6 are broken and converted back to the starting monomer, ε-caprolactam. Once purified, the monomer is polymerized again to a high-quality PA6. This is the advantage in respect to mechanical recycling, which implies a product downgrading in respect to the starting material. In general, hydrolytic depolymerization of PA6 involves the treatment of the molten polymer with water at temperatures in the range of 200–450 °C and pressure of 0.5–20 bar [3,4].

In most cases, PA6 is just an ingredient of the material that is fed to the depolymerization reaction, since the majority of post-consumer waste derive from old carpets or old fishing nets, which contain many other additives besides PA6.

For example, the typical broadloom carpet is composed of about 30% PA6 face fiber, about 10% of polypropylene as primary and secondary backings, 9% styrene-butadiene latex adhesive and about 35% calcium carbonate as filler. When, at the product end of life, the primary backing is detached from the secondary backing to recycle PA6, part of the additive (mainly the adhesive) remains on the product [5]. To give other examples, in the textiles used for high performance sportswear and activewear, PA6 is coupled with Spandex to impart higher elasticity to the yarn [6]. In other applications linked to engineering plastics, PA6 is compounded with rubbers to guarantee higher toughness and impact strength [7,8].

If those PA6 post-consumer wastes undergo the hard depolymerization conditions described previously, the reaction results in both the starting monomer ε-caprolactam, and in other substances. Those substances could be toxic, for example, the aromatic amines (e.g., aniline), used in isocyanates (principally MDI) as precursor to urethane elastomers, that are carcinogenic [9].

The above considerations highlight the need to study new materials that, added to PA6, confer to the polymer rubber-like or elastic properties, overcoming the described problems. As a whole, this added material should maximize the PA6 content, then mimic the properties of rubber. It should be amorphous and have high elongation at break and, to be a good hot melt for PA6, it needs to have a low softening point and low melting temperature [10]; for example, the commercially available hot melt adhesives for PA6 are characterized by a melting temperature in the range of 110–150 °C.

The idea for this study is to use PA6 blends or copolymers, since it is well-known that the melting temperature and mechanical properties of copolymers can vary according to the type and ratios of the considered main building blocks [11].

A polymer blend is similar to a metal alloy: two or more polymers are combined to create a new material with enhanced physical and chemical properties. Three different classes exist: 1. homogeneous polymer blend: the building blocks have similar chemical structures, yielding a homogeneous structure blend; 2. compatible polymer blends: the constituent materials are immiscible, but due to strong interaction the blend has uniform properties; 3. heterogeneous polymer blend: the polymers coexist in the blend separately and different Tg values are observed. Scope of blending with other polymers is to overcome some drawbacks of polyamide materials. Indeed, despite having good heat resistance and high mechanical strength, polyamides tend to absorb water and have insufficient impact resistance and notch sensitivity. Therefore, blending polyamides with other polymers can lead to better impact resistance, flexibility and processability properties alongside with minor water absorption [12].

In addition, copolymerization is studied to prepare polymers with desired properties. Changing the comonomers types and their relative ratio gives the ability to tune the properties as desired. Tseng et al. synthesized a series of biobased co-polyamide (co-PAs), for example, PA56/PA66 [13]. Other studies focus on the copolymerization of PA11/PA12 [14] and Zielbermann et al. prepared and characterized a random copolymer of PA6 and PA6.9 [15].

The copolymer chosen for this investigation was the one of PA6 and PA6.9; since it meets the requirement of maximizing the PA6 content with the additive as desired, additionally, the PA6.9 component is made of hexamethylenediamine and a potentially biobased monomer (azelaic acid) [16].

Indeed, the development of monomers derived from biobased has been boosted in recent years due to the depletion of fossil oil resources and the increasing research in microbiology and germ technology.

Vegetable oils represent one of the most promising sources for renewable chemicals used to produce biopolymers. One example is castor oil from Ricinus communis, that consists mainly of the C18 fatty acid ricinoleic acid, a precursor of 11-aminoundecanoic acid, the monomer of polyamide 11 (PA11) and is also a precursor of sebacic acid, used, for example, in polyamide 6.10 (PA6.10) [17].

Moreover, diamine raw materials are increasingly substituted by biobased products deriving from enzyme reaction. In this framework, cadaverine (1.5-pentanediamine) can be produced from a renewable biomass and is a raw material for synthesizing bio-polyamides such as polyamide 5.4 (PA5.4), polyamide 5.6 (PA5.6) and polyamide 5.10 (PA5.10) (succinic acid/adipic acid/sebacic acid and cadaverine) [18]. In addition, polyamide 6.9 is well inserted in this context, considering its potential biobased monomers.

Considering the thermic behavior, past studies showed that when two polymers in the same family of the polyamides are mixed, blended and kept molten for a certain length of time, the melting temperature of the resulting copolymer decreases over time, while it is moving from block to random [19]. In this specific case, mixing 50% of PA6.9 with 50% of PA6 and keeping the molten mixture at 288 °C for a fixed time, the melting temperature has been shown to decrease over time, reaching a temperature of 135 °C after 12 h [20].

The low melting temperature of the copolymer PA6.9-ran-PA6 (from now on PA6.9-6) could be promising in its use as the needed additive of PA6. However, the long timing and high temperatures required for its production do not fully comply with an industrial process. Moreover, in addition to the melting temperature, the other properties of the resulting material were not characterized in the above-mentioned work [20].

Supported by the previous studies, the present work is aimed at a deeper characterization of the copolymer PA6.9-6 and at the investigation of its possible production processes, comparing a blending and a copolymerization approach.

## 2. Materials and Methods

### 2.1. Materials

Hexamethylene diamine (from now on HMDA) provided by Arpadis (Antwerp, Belgium) azelaic acid (AZA) from Merck (Darmstadt, Germany) and from Novamont S.p.A (Novara, Italy), caprolactam (CPL) from Basf (Ludwigshafen am Rhein, Germany). 

### 2.2. Synthesis

To produce the blended polymer, PA6 and PA6.9 are first polymerized, then mixed and molten considering different ratios of the polymers. In case of the copolymer produced by direct polymerization of the three starting monomers, just one polymerization cycle is required.

For the sake of simplicity, only the PA6 and PA6.9 polymerization cycle will be described, while the others can easily be derived accordingly.

The pilot plant used for the synthesis is divided in three units (Figure 1):homogenizer: a jacketed vessel, heated with hot water (typically max 90 °C);autoclave: a jacketed vessel, heated with thermal oil (typically max 280 °C);a system to produce the chips, made by an extrusion pump, a spinneret and a cutter.

The polymerization of PA6 is carried out by charging the caprolactam monomer in the homogenizer to 80–90 °C with some water, typically 2–5%, necessary for initially opening the ring. After transferring the molten mixture in the autoclave, the polyaddition and the polycondensation reactions start and the polymerization is managed in a 4-phase cycle for a total time of approximately 5–6 h.

Temperatures are always kept between 200–250 °C, while pressure is changing from 10 bar to 200 mbar in the different phases of the cycle. Initially, there is a pressurization phase, then pressure is kept constant by regulating the vapor vent valve, then it is decreased down to atmospheric pressure, and finally, a vacuum phase is responsible for the final quick increase in the polymer molecular weight. The viscosity of the mass is controlled through the power absorption of the stirrer and, when it is high enough, the polymer mass is pushed to the extrusion pump and forced through the spinneret, a subsequent cold-water tub and a cutter, where the polymer chips are obtained.

The conversion of the caprolactam monomer to PA6 reached a level around 90%, and for this reason, an extraction with hot water is done, to remove the residual monomer or low molecular weight oligomers from the chips [21,22]. Then, a final drying step under nitrogen is realized to reach polymer chips with moisture values around 0.05–0.1%.

In the case of PA6.9, the two monomers, HMDA and AZA, need to be in a molar ratio 1:1. This is ensured by the preparation of a salt between the two monomers, in the homogenizer (this salt will be called AAH salt). Initially, the water is charged, followed by HMDA (water soluble) and then by AZA (low solubility), with an initial water amount sufficient to solubilize the salt (about 50% solubility at 80–90 °C). After about 30 min, the solution in the homogenizer is transferred to the autoclave, where polymerization is always managed in a 4-phase cycle with the same procedure and steps described to produce PA6. The final vacuum phase, with the water removal, pushes, in this case, the polymerization reaction of PA6.9 to the full conversion of the two monomers to 100%.

In addition, the 4-phase cycle for PA6.9 is completed in a total time of approximately 5–6 h.

The copolymerized PA6.9-6 is realized with the polymerization cycle described above mixing HMDA, AZA and CPL in the correct ratios to reach the desired final composition.

In addition, when dealing with PA6.9-6 the reaction conversion is not completed for the monomer caprolactam, so for the copolymers it is again carried out an extraction with hot water.

The blended polymer is realized by mixing different ratios of chips of PA6 and PA6.9, melting the mixed chips through an extruder (Sandretto, Torino, Italy) and pelletizing the as-obtained mixed polymer in chips.

The length of the extruder is 600 mm (28 mm diameter), the speed of extrusion is 10 mm/s and the temperature is 280 °C.

### 2.3. Characterization Techniques

#### 2.3.1. Thermic Characterization

The Differential Scanning Calorimetry (DSC) tests were carried out using a TA Instrument DSC Q20 (TA Instruments, New Castle, DE, USA). The measurements were performed on a sample of about 10 mg, under nitrogen flow of 100 mL/min. The sample is first heated from 20 to 280 °C at a rate of 20 °C/min, then cooled to 0 °C and heated again to 280 °C.

The samples’ Heat Deflection Temperature (HDT) was determined via a Instron HV 3S (Instron, Torino, Italy), following ISO 75.

#### 2.3.2. X-ray Diffraction

Powder XRD patterns were collected on a Bruker D8 Advance capillary diffractometer (Bruker, Massachusetts, USA) equipped with a Cu Kα source (λ = 1.5418 Å, 40 kV, 40 mA) and considering a Bragg-Brentano geometry (0.1° steps from 5–50°, ambient temperature and in rotation). The diffraction patterns were fitted by a least-square procedure, to quantify the contributions of the different phases to the final crystallinity, elaborated by Ho et al. for PA6 [23] and by Murase et al. for PA6.9 [24].

#### 2.3.3. Mechanical Characterization

The copolymer chips were remelted via an injection molding machine (Sandretto, Torino, Italy), in which through a screw extruder set at a temperature around 240 °C, the molten material is injected at the high pressure of 600 bar to fill the mold and set at a temperature around 50 °C. The mold cavity has the shape to produce the lab test specimens of dumbbell-shape (1 A type) and of Izod impact specimen shape.

The dumbbell-shaped specimens have been tested via a Dynamometer Instron 34 TM-10 (following ISO 527, Instron, Torino, Italy) and the Izod impact test is performed via a Instron 9050 (following ISO 180, Instron, Torino, Italy).

#### 2.3.4. Nuclear Magnetic Resonance

The NMR measurements were performed with a Bruker-Avance 400 MHz spectrometer (Bruker, Massachusetts, USA).

Operating with a stationary magnetic field at a strength of 9.4 T and equipped with a 5 mm BBI probe capable of providing longitudinal pulsed-field gradients up to 53 cm^−1^.

The polymer is dissolved in deuterated formic acid (Sigma-Aldrich, St. Louis, MO, USA, DCOOD, about 70 mg in 800 µL) and both the spectra of ^1^H-NMR and ^13^C-NMR have been analyzed. Another possible preparation method (referred as CD_3_COOD/HCOOH) consists of the dissolution of the sample in formic acid (Sigma-Aldrich, St. Louis, MO, USA, about 30 mg in 200 µL) with a dilution of deuterated acetic acid (Sigma-Aldrich, Missouri, USA, 600 µL).

## 3. Results and Discussion

### 3.1. PA6.9/PA6 Blending

The first material studied has been the blended polymer: PA6 and PA6.9, which have been polymerized according to the procedure described in the previous section, then the chips have been mixed in different ratios, fed into the hopper of an extruder, molten at about 280 °C and extruded again.

Considering, for example, the case of 40% PA6.9 and 60% PA6 by weight, the copolymer melting temperatures have been characterized via DSC.

Figure 2 shows the reported the curves of the two starting polymers and the final blend.

It can be easily observed that the blend melting temperature is in the same range of the starting polymers (215–220 °C). This in an indication that the material is still separated in the two main components and for this reason, the melting temperature did not change. Keeping the polymer molten at a high temperature (around 280 °C) for a longer time (about 12 h) would have provided enough time for amide interchange reactions necessary for the conversion of the polymer from block to random and the subsequent reduction of the material melting temperature [20,25].

### 3.2. PA6.9-6 Copolymerization

Here, a different approach has been tried, which is the direct copolymerization of the three starting monomers, since it could be less time and energy demanding. This approach has been patented for PA6 and PA6.6 copolymers, but only with a material characterization for small amounts of PA6.6 (up to 10%) with respect to the PA6 component [26]. Other studies demonstrated the feasibility of another copolymerization approach, based on the addition of CPL to PA6.6 with subsequent polymerization, keeping also in this case a low PA6.6 content [27,28]. The polymer obtained in those examples was established to be random, showing the expected decrease in melting temperature.

Supported by the preliminary results found in literature, different ratios of AZA/HMDA and CPL have been copolymerized, washed and dried as described in Section 2.2, to cover the entire spectrum of different PA6.9-6 compositions (80-20, 60-40, 50-50, etc.).

The material has been characterized in terms of viscosity, terminal groups and residual monomer and, more interestingly for the present work, the thermic behavior was studied.

The DSC curve of the copolymerized PA6.9-6 60-40 obtained by direct polycondensation of the three monomers in maximum 6 h. Figure 3a shows a melting temperature around 144 °C, much lower than the corresponding polymers and their blend in the same ratio obtained by mixing and extruding the two individual polymers.

The melting temperature is decreasing with the increase in AAH salt component, up to a value of 135 °C when the PA6.9 and PA6 content are 50–50% in weight (Figure 2b). This melting temperature agrees with the value obtained blending and keeping the two polymers at 288 °C after 13 h. In that case, it was stated that the material obtained was a random copolymer [20].

The insight into this copolymer structure was done by NMR, since it is well-reported to be a good technique for the evaluation of the degree of randomness of a copolymer [28,29,30]. As a first check, the PA6.9 NMR spectrum was analyzed and as reported in Figure 4, the measured chemical shifts are in good agreement with the expectations in both spectra. Moving to the copolymerized PA6.9-6 and considering the repeating units of PA6 and of PA6.9 (AAH salt, which is also formed in the copolymerization case), the four possible configurations are -6-69-6-69, -6-69-69-6, -6-6-69-69, -69-6-6-69. In the four configurations, the chemical surrounding of each carbon atom is different; hence, NMR should be capable of detecting those differences.

A sample of copolymerized PA6.9-6 60-40 has been analyzed via NMR and the resulting spectrum is shown in Figure 5.

Each peak of the spectrum of the pure PA6.9 is split in four peaks, given by the four different chemical surroundings of the sp^3^ carbons in the different configurations (the slightly higher chemical shift is due to the different solvent used in this case with respect to Figure 4). The intensities of the four signals are similar, in agreement with the 60-40 ratio of the starting monomers. This result gives a clear indication that the copolymerization process results in a random copolymer, with an easy modulation of the melting temperature according to the ratio of the starting monomers.

The randomness of the polymers also ensures the uniformity of the properties across the whole produced material, which has been investigated for the different monomer ratios.

First, the samples crystallinity has been studied via XRD measurements, and the resulting spectra are reported in Figure 6.

The measured crystallinity, considered as the ratio of the crystalline phases with respect to the whole area (given by the sum of amorphous and crystalline phases) is 32% for PA6, 31% for PA6.9, 24% for PA6.9-6 80-20 and 22% for PA6.9-6 60-40. As expected, the copolymer shows a decrease in the crystallinity, as can be easily deduced also considering the high transparency of the copolymer samples. In addition, looking at the deconvolution of XRD spectra into the contribution of the different crystalline phases, the crystalline structure and the contribution of each phase is different for each polymer.

Regarding the mechanical characterization, lab test specimens of the dumbbell-shape and the Izod impact shape specimens have been printed as described in Section 2.3.2 for the different copolymer compositions. The tests results (with a repetition of each measurement over five samples) are reported in Table 1: moreover, a comparison of three different stress-strain curves is reported in Figure 7.

As expected, the blended polymer showed properties similar to the starting polymers. Additionally, the copolymerized PA6.9-6 shows a high impact resistance, as the unnotched Izod measurement resulted in energy absorption without breakage. The material is also characterized by high elongation without breakage (as highlighted in Figure 7) and by a low softening temperature. 

Those features agree with the rubber-like properties, which are required for the applications mentioned in the introduction.

## 4. Conclusions and Future Work

The present work showed how to produce a random copolymer PA6.9-6. Tuning the relative concentrations of the starting monomers allows for modification to the desired mechanical and thermal properties, according to the final application. The synthetized copolymers can be used as an additive in polyamide matrix (such as PA6); therefore, the concentration of polyamide 6 is increased and hence the yield of caprolactam may be higher. Furthermore, the additive does not contain materials (such as in polyurethane elastomers or rubbers) that, once in depolymerization, can release carcinogenic aromatic amines.

Indeed, PA6.9-6 has shown rubber-like properties: it is highly amorphous, it is characterized by a low glass transition temperature, high elongation without breakage and high impact resistance. The melting temperature varies from 140 °C to 220 °C depending on the relative PA6.9 and PA6 content.

The copolymer can be used as a hot melt. There are ongoing tests regarding the use of the powder PA 6.9-6 50-50 as a hot melt and its performance. Other possible applications deal with the use of the copolymer in compounding to impart rubber-like properties to PA6 or to improve PA6 yarn elastic properties.

## Figures and Tables

**Figure 1 polymers-14-04115-f001:**
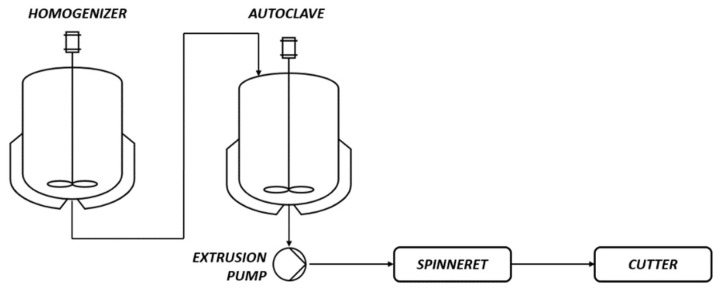
Basic scheme of the used polymerization plant.

**Figure 2 polymers-14-04115-f002:**
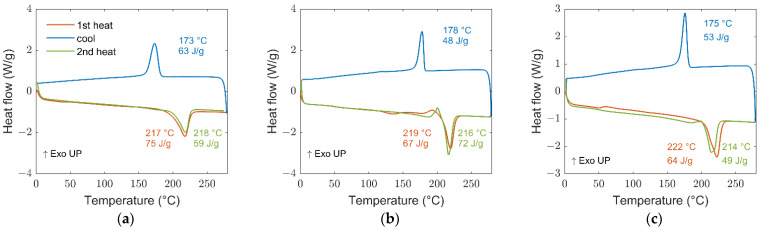
DSC curves of PA6 (**a**), PA6.9 (**b**) and blended polymer of 40% PA6.9, 60% PA6 (**c**).

**Figure 3 polymers-14-04115-f003:**
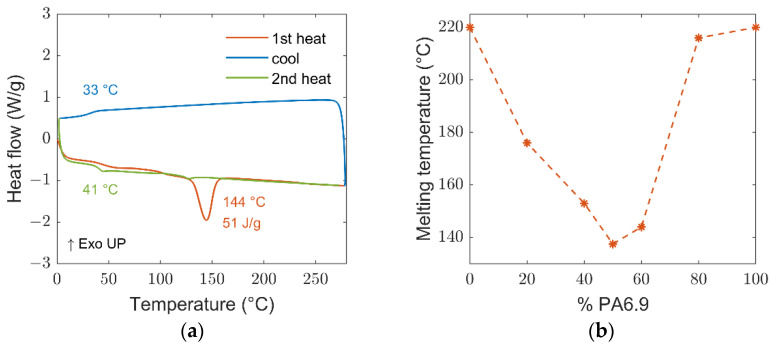
DSC curves of PA6.9-6 60-40 with indication of melting and glass transition temperatures (**a**), melting temperatures for different monomers ratio (**b**).

**Figure 4 polymers-14-04115-f004:**
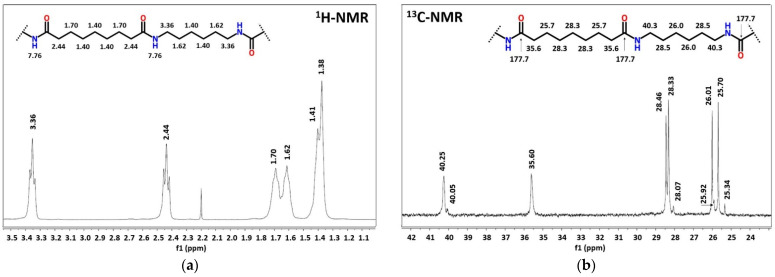
NMR spectrum of PA6.9 (in DCOOD) with interpretation of chemical shifts, ^1^H-NMR spectrum (**a**) and ^13^C-NMR spectrum (**b**).

**Figure 5 polymers-14-04115-f005:**
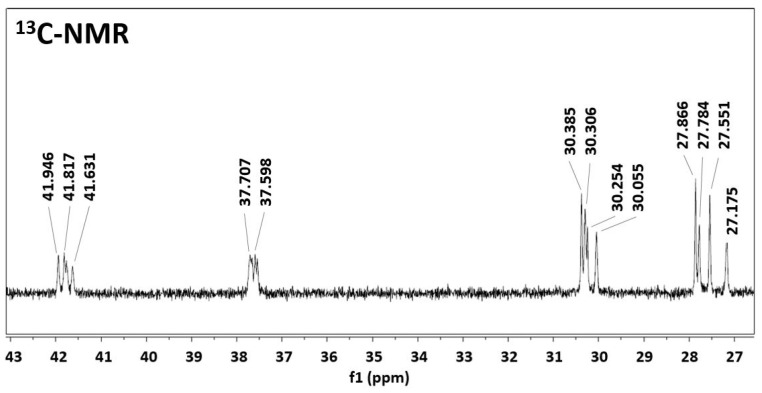
NMR spectrum of PA6.9-6 60-40 (in CD_3_COOD/HCOOH).

**Figure 6 polymers-14-04115-f006:**
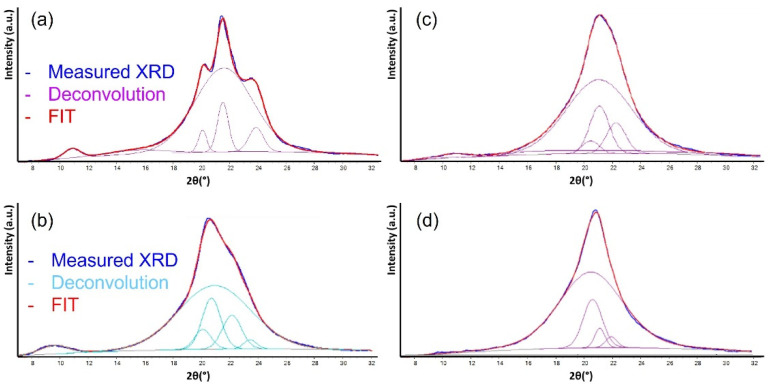
XRD spectra with deconvolution of the different crystalline and amorphous phases for PA6 (**a**), PA6.9 (**b**), PA 6.9-6 80-20 (**c**), PA6.9-6 60-40 (**d**).

**Figure 7 polymers-14-04115-f007:**
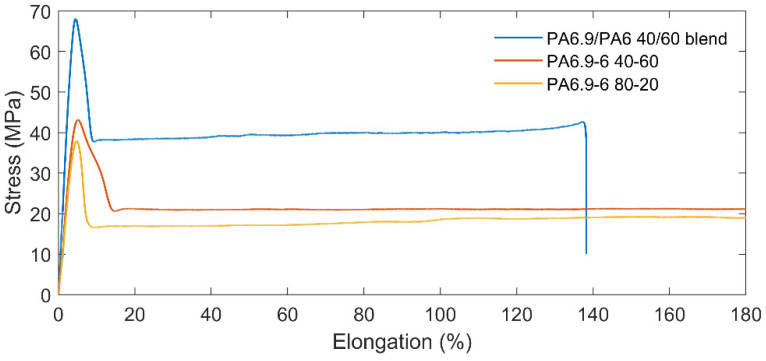
Examples of stress-strain curves for blended PA6/PA6.9 and for two different copolymers.

**Table 1 polymers-14-04115-t001:** Properties of pure polymers, copolymers (80-20, 60-40, 40-60) and blend (40-60).

	PA6.9	PA6.9-6 80-20	PA6.9-6 60-40	PA6.9-6 40-60	PA6.9/PA6 40/60 (Blend)	PA6
Melting T [°C]	220 ± 2	216 ± 2	144 ± 2	156 ± 2	218 ± 2	220 ± 2
Relative viscosity	3.1 ± 0.1	3.3 ± 0.1	3.0 ± 0.1	3.2 ± 0.1	3.1 ± 0.1	3.4 ± 0.1
Tensile strength [MPa]	56.4 ± 0.3	37 ± 1	35 ± 1	43 ± 1	67 ± 1	68 ± 1
Strain at yield [%]	4.6 ± 0.1	4.6 ± 0.1	4.5 ± 0.2	4.5 ± 0.1	4.2 ± 0.1	4.6 ± 0.1
Elongation at break [%]	80 ± 9	>180 (N)	>180 (N)	>180 (N)	124 ± 15	82 ± 12
Notched Izod [J/m]	43 ± 5	33 ± 6	42 ± 3	51 ± 3	38 ± 4	50 ± 4
Unnotched Izod [kJ/m]	1.8 ± 0.1	1.2 ± 0.1 (N)	1.3 ± 0.1 (N)	1.5 ± 0.1 (N)	2.1 ± 0.1	2.3 ± 0.1
HDT @ 1.8 MPa [°C]	45.4 ± 0.2	28.9 ± 0.1	29.2 ± 0.1	30.4 ± 0.2	45.5 ± 0.2	45.3 ± 0.2

(N) Indicates that the sample did not break.

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
