# Peer review of "Comparison of the Properties of a Random Copolymer and a Molten Blend PA6/PA6.9"

_polymers, 2022, doi:10.3390/polym14194115_

Round 1

Reviewer 1 Report

The authors describe the comparison of the properties of PA6/PA6.9 with a mixed molten blend or a random copolymer. The advantage is the chemical recyclability owing to the high PA6 content. Although the materials are potentially useful in applications, I cannot recommend the manuscript for publication. There are numerous problems with lacking information and in the presentation of results, as listed below.

1. In the Introduction, the authors should introduce more information about the progress of biobased PA, such as PA56.

2. What is the molecular weight and viscosity of as-synthesized PA? and the authors should analyze the effect of the ratio of PA6 and PA6.9. If the molecular weights of the same copolymer synthesized are different, does this have an impact on the final test results?

3. Why the copolymer is block and random? More information should be provided to these results.

4. The sequence distribution, crystal morphology, crystal forms, and melting and crystallization behavior of copolymers should be systematically investigated by quantitative. Although the DSC curves of materials were provided, the thermal behavior was lack of comparative analysis, and relation between structure and performance.

5. The stress-strain cures should be supplemented and shown in the figures. The error bar should be supplemented in Table 1.

6. The mechanism of rubber-like properties were not clear, why the materials are high elongation without breakage and by a low softening temperature.

7. How biodegradable is these materials, the relative experimental studies should be provided.

8. What is the spinning properties of these materials?  

9. Why the melt blended PA6/PA6.9 is copolymer? Are the two different macromolecules sufficiently chemical bonded? Detail of the production of the blended polymer copolymer should be described.

10. The density of the hydrogen bond has a great influence on the performance of nylon, how is the density of the hydrogen bond measured in this paper?

Reviewer 2 Report

Thank you for the opportunity to review the article entitled: Comparison of the properties of a random copolymer and a 2 molten blend PA6/PA6.9. Below, I present my criticisms of the presented manuscript.

The abstract should be rewritten. It should not contain as much information regarding literature reports. However, it lacks the purpose of the research, short description of the results and conclusions.

The literature section on the synthesis of copolymers and PA mixtures produced in the work should be expanded.

Nylon is a trade name and should not be used in this manuscript.

The nomenclature of PA6.9-6 copolymers is colloquial and inconsistent with the applicable rules for the nomenclature of copolymers.

The description of the method of obtaining PA blends should be expanded and included in the methodology.

Why in the case of blends, the obtained materials are called copolymers? Has it been proven that there is a copolymerization reaction in this case? Please check the impact unit.

On the thermogramming, please mark the direction of exo or endo on the axis.

 NMR spectra not readable. I suggest you improve the quality of the numerical values.

For comparison, NMR and XRD tests for PA blends should be performed.

To determine the rubber-like properties, a study of the glass transition temperature and modulus of elasticity should be performed.

Conclusions not always adequate to the obtained results. For example: The studied copolymers can be used as additive for Nylon 6, since they maximize the Nylon 6 content in the final material and in the depolymerization process no toxic substances are released… - This has not been studied in this work.

Round 2

Reviewer 1 Report

This manuscript can be accepted as it is